# Prediction of inter-chain distance maps of protein complexes with 2D attention-based deep neural networks

Zhiye Guo [1], Jian Liu[1], Jeffrey Skolnick[2] & Jianlin Cheng [1] ✉

Residue-residue distance information is useful for predicting tertiary structures of protein monomers or quaternary structures of protein complexes. Many deep learning methods have been developed to predict intra-chain residue-residue distances of monomers accurately, but few methods can accurately predict inter-chain residue-residue distances of complexes. We develop a deep learning method CDPred (i.e., Complex Distance Prediction) based on the 2D attention-powered residual network to address the gap. Tested on two homodimer datasets, CDPred achieves the precision of 60.94% and 42.93% for top L/5 inter-chain contact predictions (L: length of the monomer in homodimer), respectively, substantially higher than DeepHomo's 37.40% and 23.08% and GLINTER's 48.09% and 36.74%. Tested on the two heterodimer datasets, the top Ls/5 inter-chain contact prediction precision (Ls: length of the shorter monomer in heterodimer) of CDPred is 47.59% and 22.87% respectively, surpassing GLINTER's 23.24% and 13.49%. Moreover, the prediction of CDPred is complementary with that of AlphaFold2-multimer.

Proteins are a key building block of life. The function of a protein is largely determined by its three-dimensional structure[1]. Sometimes single-chain proteins (monomers) can perform certain functions, while the structures of most individual proteins interact to form multi-chain complex structures (multimers) to carry out their biological function[2]. Therefore, modeling the three-dimensional structure of both monomers and protein complexes is crucial for studying protein function.

Deep learning has been applied to advance the prediction of the tertiary structures of monomers since 2012[3]. Over a decade, many deep learning methods were developed to predict intra-chain residue-residue contact maps or distance maps of monomers[4-8], which were used by contact/distance-based modeling methods such as CONFOLD[9] and Rosetta[10] to build their tertiary structures. Extensive studies[9,11-13] have shown that if a sufficiently accurate intra-chain distance map is predicted, then the protein's tertiary structure can be accurately constructed. Most recently, AlphaFold2[14] uses an end-to-end deep learning method to predict both tertiary structures and residue-residue distances of monomers, achieved a very high average accuracy (~90 Global Distance Test (GDT-TS) score[15] in the 14th Critical

Assessment of Techniques for Protein Structure Prediction (CASP14) in 2020. Recently, AlphaFold2 was extended to AlphaFold-multimer[16] and AF2Complex[17] to improve the prediction of quaternary structures of multimers.

Following the deep learning revolution in the prediction of intra-chain residue-residue distances and tertiary structures, recently some deep learning methods were developed to predict the inter-chain residue-residue contact map of homodimers and/or heterodimers, such as ComplexContact[18], DeepHomo[19], DRcon[20], and GLINTER[21] that predicts the contact map for both homodimers and heterodimers using as input a graph representation of protein monomer structure and the row attention maps generated from multiple sequence alignments (MSAs) by the MSA transformer[22]. The attention map calculated by the MSA transformer is a kind of residue-residue co-evolutionary feature extracted from MSAs. It has been automatically trained on millions of MSAs to capture the co-evolutionary information across many diverse protein families during its unsupervised pretraining. Despite the significant progress, the accuracy of inter-chain contact prediction is still much lower than that of intra-chain contact/distance

[1]Department of Electrical Engineering and Computer Science, University of Missouri, Columbia, MO 65211, USA. [2]School of Biological Sciences, Georgia Institute of Technology, Atlanta, GA 30332-200, USA. ✉e-mail: chengji@missouri.edu

prediction, which calls for the development of more methods to tackle this problem.

In this work, we develop a protein complex distance prediction method (CDPred) based on a deep learning architecture combining the strengths of the deep residual network[23], a channel-wise attention mechanism, and a spatial-wise attention mechanism to predict the inter-chain distance maps of both homodimers and heterodimers. As in GLINTER, the attention map of the MSA generated by the MSA transformer is used as one input for CDPred. The predicted distance map for monomers in dimers is used as another input feature. Different from the existing deep learning methods, CDPred predicts inter-chain distances rather than binary inter-chain contacts (contact or no contact) that the current methods, such as DeepHomo and GLINTER predict. We test the CDPred rigorously on two homodimer test datasets and two heterodimer test datasets. For these datasets, CDPred yields much higher accuracy than DeepHomo and GLINTER.

## Results

### Evaluation of inter-chain contact prediction for homodimers

We compare CDPred with DNCON2_inter[24], DeepComplex[25], DeepHomo, and GLINTER on the HomoTest1 homodimer test dataset with the results shown in Table 1. The input tertiary structures for all three methods are predicted structures corresponding to the unbound monomer structures. The DNCON2_inter is run with the recommended parameters. The DeepComplex web server is used to get its prediction results. The results of DeepHomo are obtained from its publication. Three versions of CDPred are tested. The first version (CDPred_BFD) uses the MSAs generated from the BFD database as input. The second version (CDPred_Uniclust) uses the MSAs generated from the Uniclust30 database as input. The third version (CDPred) uses the average of the distance maps predicted by CDPred_BFD and CDPred_Uniclust as the prediction. Because DeepHomo and GLINTER predict binary inter-chain contacts at an 8 Å threshold instead of distances, we convert the inter-chain distance predictions of CDPred, CDPred_BFD, and CDPred_Uniclust into binary contact predictions for comparison. The definition of inter-chain contact is the same as GLINTER and DeepHomo, i.e., a pair of inter-chain residues is considered to be in contact if the distance between their two closest heavy atoms is less than 8 Å. This definition is used to evaluate all the inter-chain contact predictions in this work.

CDPred achieves the highest contact prediction precision across the board among all the methods. For instance, CDPred has a top L/5 contact prediction precision of 60.94%, which is 50.34% percentage points higher than DNCON2_inter, 9.64% percentage points higher than DeepComplex, 23.54% percentage points higher than DeepHomo, and 12.85% percentage points higher than GLINTER. CDPred performs better than DNCON2_inter, DeepComplex, DeepHomo, and GLINTER also in terms of Accuracy Rate and AUC score and second best in terms of Accuracy Order. According to almost all the evaluation metrics,

CDPred performs better than both CDPred_BFD and CDPred_Uniclust, indicating that averaging the distance predictions made from the two kinds of MSAs can improve the prediction accuracy.

We also compared the methods above on the HomoTest2 homodimer test dataset (Table 2). CDPred performs best in terms of all the evaluation metrics. Combining the predictions of CDPred from two kinds of MSAs improves the prediction accuracy.

### The impact of MSA depth on the accuracy of inter-chain contact prediction for homodimers

Section 2.1 shows that two different MSAs (BFD and Uniclust) lead to different prediction accuracy for CDPred_BFD and CDPred_Uniclust, and CDPred that averages the two contact maps predicted from the two MSAs yields the best result. Here, we investigate how the depth of MSAs and a direct combination of the two MSAs may affect prediction accuracy. Supplementary Tables 1, 2 reports the number of sequences and the number of effective sequences (Neff)[26] for each dimer in HomoTest1 and HomoTest2 as well as the top L/2 contact prediction precision of CDPred_BFD, CDPred_Uniclust, CDPred, and CDPred_ComMSA that uses the simple combination of the BFD MSA and Uniclust MSA as input. Neff weights similar sequences in MSA less in counting the number of sequences and is widely used to measure the depth of MSA.

The Neff and contact prediction precision for CDPred_BFD and CDPred_Uniclust vary from target to target. The Pearson correlation coefficient between the difference of Neff and the difference of the top L/2 precision for CDPred_BFD and CDPred_Uniclust is 0.31 and 0.67 on HomoTest1 and HomoTest2, respectively, indicating that the depth of MSA has some positive impact on the contact prediction precision. CDPred_ComMSA, which combines the two MSAs to generate a deeper MSA as input, performs better than both CDPred_BFD and CDPred_Uniclust on HomoTest1 and better than CDPred_BFD on HomoTest2, suggesting that directly combining two MSAs can be beneficial.

CDPred still performs slightly better than CDPred_ComMSA in terms of top L/2 prediction precision on both datasets (55.19 versus 55.13% on HomoTest1 and 38.14 versus 36.14% on HomoTest2), indicating that averaging the distance maps predicted from the two MSAs is more effective than simply combining the two MSAs as input.

### Evaluation of inter-chain contact prediction for heterodimers

We compare CDPred and a state-of-the-art heterodimer contact predictor GLINTER on both HeteroTest1 and HeteroTest2 heterodimer test datasets (see results in Tables 3, 4, respectively). The input tertiary structures of monomers used by both methods are predicted by AlphaFold2. We use two different orders of monomer A and monomer B (AB and BA) in each heterodimer to generate input features for CDPred to make predictions. The average of the outputs of the two orders is used as the final prediction. The process of averaging the two outputs is shown in Supplementary Figure 1. The inter-chain part of the

**Table 1 | The precision of top 5, top 10, top L/10, top L/5, and top L contact predictions, accuracy order (AccOrder), accuracy rate (AccRate), and AUC (area under receiver operating characteristic curve) score on the HomoTest1 test dataset for DNCON2_inter, DeepComplex, DeepHomo, GLINTER, and three versions of CDPred**

| Predictors | top 5 | top 10 | top L/10 | top L/5 | top L/2 | top L | AccOrder (‰) | AccRate (%) | AUC |
|---|---|---|---|---|---|---|---|---|---|
| DNCON2_inter | 10.71 | 10.00 | 11.39 | 10.60 | 7.04 | 3.84 | 642.92 | 14.29 | 0.51 |
| DeepComplex | 57.86 | 56.07 | 54.81 | 51.30 | 41.29 | 34.88 | 40.29 | 71.43 | 0.84 |
| DeepHomo | - | 52.50 | 43.20 | 37.40 | 28.20 | - | **2.10** | 67.90 | - |
| GLINTER | 54.81 | 54.07 | 50.54 | 48.09 | 41.90 | 34.91 | - | - | 0.88 |
| CDPred_BFD | 63.57 | 62.50 | 61.24 | 58.26 | 52.78 | 47.08 | 8.61 | 75.00 | **0.90** |
| CDPred_Uniclust | 65.71 | 61.79 | 60.53 | 58.18 | 54.14 | 47.91 | 11.12 | 75.00 | 0.89 |
| CDPred | **66.43** | **65.71** | **63.14** | **60.94** | **55.19** | **49.01** | 7.76 | **75.00** | 0.89 |

L: sequence length of a monomer in a homodimer. Bold numbers denote the best results. AccOrder is the rank of the first correct contact prediction divided by the number of residues of a dimer. The smaller is AccOrder, the better is the performance. AccRate is the percentage of dimers for which at least one of the top 10 inter-chain contact predictions is correct.

**Table 2 | The precision of top 5, top 10, top L/10, top L/5, and top L contact predictions, accuracy order, accuracy rate, and AUC score on the HomoTest2 test dataset for DeepHomo, GLINTER, and CDPred predictors**

| Predictor | top 5 | top 10 | top L/10 | top L/5 | top L/2 | top L | AccOrder (‰) | AccRate (%) | AUC |
|---|---|---|---|---|---|---|---|---|---|
| DNCON2_inter | 11.30 | 9.57 | 11.38 | 6.91 | 3.74 | 3.16 | 609.17 | 17.39 | 0.50 |
| DeepComplex | 38.26 | 35.65 | 32.47 | 29.13 | 23.49 | 19.12 | 5.40 | 52.17 | 0.72 |
| DeepHomo | - | 30.43 | 27.32 | 23.08 | - | - | - | - | - |
| GLINTER | - | 43.04 | 40.18 | 36.74 | - | - | - | - | - |
| CDPred_BFD | 43.48 | 41.74 | 42.24 | 40.01 | 35.92 | 33.80 | 3.41 | 60.87 | 0.89 |
| CDPred_Uniclust | 48.70 | 45.22 | 43.32 | 39.64 | 37.46 | 32.32 | 1.32 | 65.22 | 0.86 |
| CDPred | **48.70** | **48.26** | **47.11** | **42.93** | **38.14** | **34.52** | **1.25** | **66.96** | **0.89** |

Bold numbers denote the highest precision.

**Table 3 | The evaluation of contact predictions on the HeteroTest1 test dataset for the DeepComplex, GLINTER, and CDPred**

| Predictor | top 5 | top 10 | top Ls/10 | top Ls/5 | top Ls/2 | top Ls | AccOrder(‰) | AccRate (%) | AUC |
|---|---|---|---|---|---|---|---|---|---|
| DeepComplex | 13.33 | 7.78 | 9.86 | 7.40 | 4.79 | 3.73 | **1.43** | 33.33 | 0.58 |
| GLINTER | - | 24.44 | 29.70 | 23.24 | - | - | - | - | - |
| CDPred | **55.56** | **54.44** | **51.47** | **47.59** | **38.64** | **32.73** | 16.90 | **77.78** | **0.81** |

Ls: the sequence length of the shorter monomer in a heterodimer. Bold numbers denote the best result.

**Table 4 | The evaluation of contact predictions on the HeteroTest2 test dataset for the DeepComplex, GLINTER, and CDPred**

| Predictor | top 5 | top 10 | top Ls/10 | top Ls/5 | top Ls/2 | top Ls | AccOrder(‰) | AccRate (%) | AUC |
|---|---|---|---|---|---|---|---|---|---|
| DeepComplex | 7.00 | 7.00 | 5.44 | 5.63 | 5.01 | 4.34 | 191.38 | 10.00 | 0.57 |
| GLINTER | 14.55 | 13.27 | 13.73 | 13.49 | 12.27 | 10.40 | - | - | - |
| CDPred | **23.27** | **23.82** | **23.93** | **22.87** | **20.17** | **17.51** | **62.14** | **32.73** | **0.77** |

Ls: the sequence length of the shorter monomer in a heterodimer. Bold numbers denote the best result.

BA prediction map is taken out and transposed to the same shape as its counterpart in the AB prediction map before they are averaged.

On the HeteroTest1 dataset (Table 3), CDPred achieves much better performance than GLINTER in terms of all the metrics. It is also substantially better than DeepComplex in terms of all the metrics but Accuracy Order. For instance, the top Ls/5 contact prediction precision of CDPred, 47.59% is more than twice 23.24% that of GLINTER, and 40.19% percentage points higher than DeepComplex. On the HeteroTest2 dataset (Table 4), CDPred also substantially outperforms DeepComplex and GLINTER in terms of all the metrics (contact precisions, Accuracy Order, Accurate Rate, and/or AUC).

Supplementary Tables 3, 4 compare the performance of using the two different orders of monomers as input (CDPred(A_B) and CDPred(B_A)) and averaging the outputs of the two different orders (CDPred) on the HeteroTest1 and HeteroTest2 datasets, respectively. The accuracy of CDPred(A_B) and CDPred(B_A) varies from target to target and from dataset to dataset. Sometimes the precision of the two orders can be substantially different (see Supplementary Fig. 2 for a target-by-target comparison of the precision on HeteroTest1 and HeteroTest2). However, a two-sided pairwise t-test shows that there is no significant difference between the two on average. Even though averaging the contact maps predicted in two different orders does not always yield the best accuracy, it makes the performance more stable by reducing the variance and smoothing the prediction. For instance, CDPred often delivers either the best or medium prediction accuracy in comparison with CDPred(A_B) and CDPred(B_A).

Furthermore, we divide the top L/10 contact prediction precisions for the heterodimers in the more challenging HeteroTest2 dataset into four equal intervals and plot the number of heterodimers in each interval (Fig. 1). The precision of the predictions in the four internals is bifurcated, mainly centered on a low precision interval [0–25%] and a high precision interval [75–100%]. Forty heterodimers have low

contact prediction precision in the range of 0–25%, indicating there is still a large room for improvement. One reason for the low precision is that most of the 40 heterodimers have shallow MSAs. The Pearson correlation coefficient between the logarithm of the number of effective sequences (Neff) of MSA and the top L/10 complex contact precision is 0.46, indicating a modest correlation between the two.

It is also observed that the inter-chain contact prediction accuracy for heterodimers is lower than for homodimers on average. One reason is that the MSA generation for a homodimer only needs to generate an MSA for a monomer in the homodimer, which is usually much deeper than the MSA generated for a heterodimer that requires the pairing of the related sequences in the MSAs of two different monomers in the heterodimer. Another reason is that homodimers tend to have a larger interaction interface than heterodimers on average, making the prediction easier.

**Comparison of the co-evolutionary features generated by the statistical optimization method and deep learning method**

To compare the performance of the co-evolutionary feature generated by the statistical optimization tool −CCMPred and the deep learning tool−MSA transformer, we trained two different models on the two different kinds of co-evolutionary features of the same training dataset using the same neural network architecture. One network (CDPred_PLM) is trained on the PLM co-evolutionary features generated by CCMPred. Another one (CDPred_ESM) is trained on the row attention map features generated by the MSA transformer. The precision of the top L/10 contact predictions of the two models on the four different test datasets are plotted in Fig. 2. CDPred_ESM has better performance than CDPred_PLM on all four test datasets, indicating that the co-evolutionary feature extracted automatically by the deep learning method is more informative than by the statistical

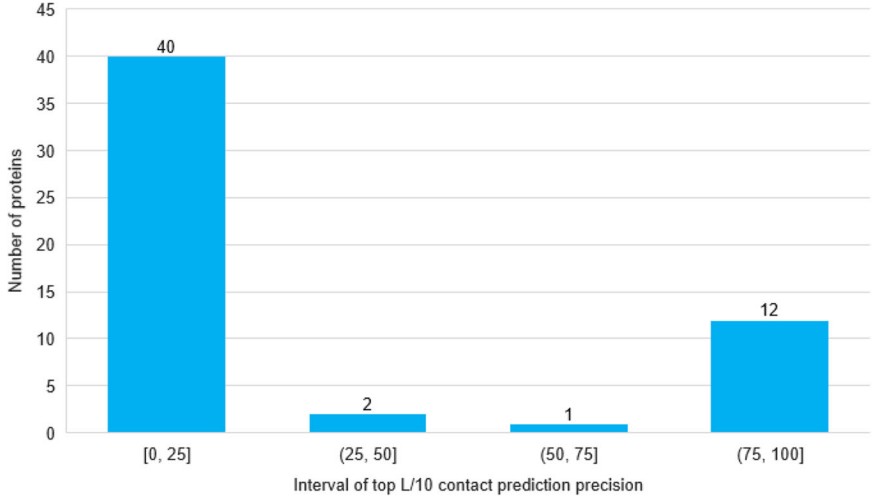

**Fig. 1 | The histogram of the precision of the top L/10 contact predictions for the heterodimers in the HeteroTest2 dataset.** The X-axis is the four precision intervals from 0 to 100%. The Y-axis is the number of heterodimers whose contact precision falls in each interval. Each interval has 40, 2, 1, and 12 heterodimers, respectively.

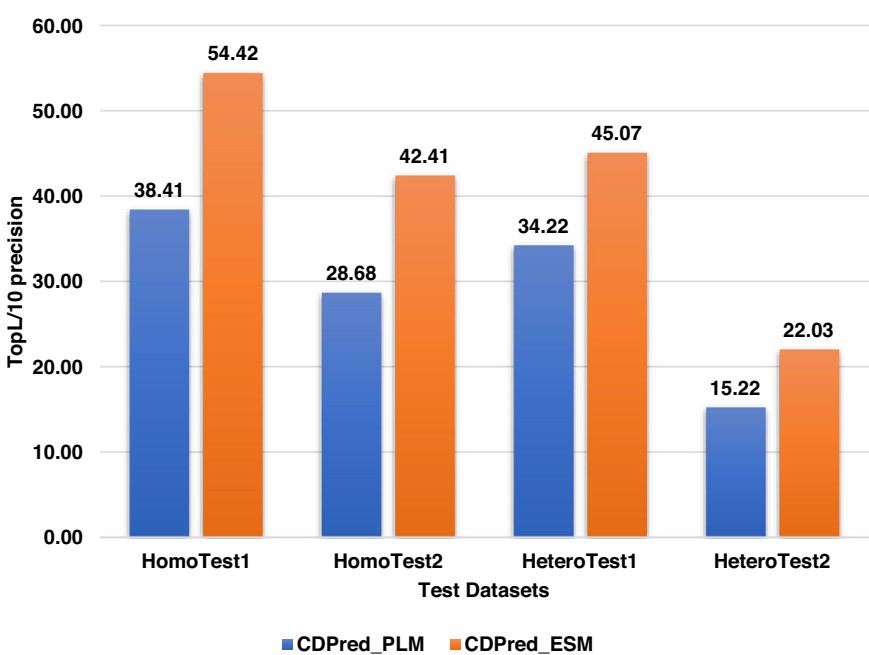

**Fig. 2 | Comparison between CDPred_PLM (blue) and CDPred_ESM (orange) on four different test datasets.** The y-axis is the top L/10 contact prediction precision, and the x-axis is the four different test datasets.

optimization method of maximizing direct co-evolutionary signals. However, combining the two kinds of co-evolutionary features yields even better results (see the results in Tables 1, 2, 3, and 4). Supplementary Figure 3 plots the top L/10 precision of CDPred_ESM against the top L/10 precision of CDPred_PLM for the homodimers in the two homodimer test datasets and the heterodimers in the two heterodimers test datasets, respectively. For 42 out of 51 homodimers and 55 out of 64 heterodimers, CDPred_ESM has higher precision than CDPred_PLM. Both CDPred_ESM and CDPred_PLM can perform better on some targets, indicate the co-evolutionary features used by the two methods have some complementarity.

**The impact of the quality of predicted tertiary structures of monomers on inter-chain distance prediction of dimers**
The quaternary structure of a protein complex depends on the tertiary structure of its monomer units. As Alphafold can predict the tertiary

structure of monomers very well, we investigated how effectively AlphaFold-predicted tertiary structures can be applied to predict inter-chain distance maps for protein complexes. The TM-scores of the predicted tertiary structure for each monomer unit of each dimer and the contact prediction precision of CDPred on the four datasets (HomoTest1, HomoTest2, HeteroTesst1, and HeteroTest2) are shown in Supplementary Tables 5, 6, 7, 8, respectively. The average TM-scores of the predicted tertiary structures for HomoTest1 and HomoTest2 are 0.95 and 0.90, for Chain A of heterodimers in HeteroTest1 and HeteroTest2 are 0.90 and 0.89, and for Chain B of heterodimers in HeteroTest1 and HeteroTest2 are 0.95 and 0.88, respectively, indicating the AlphaFold-predicted tertiary structures have high quality. The Pearson's correlation between the TM-score of the predicted tertiary structures and top L/2 contact prediction precision is 0.19. The weak correlation may be partly due to that the quality of predicted tertiary structures is high enough in general for

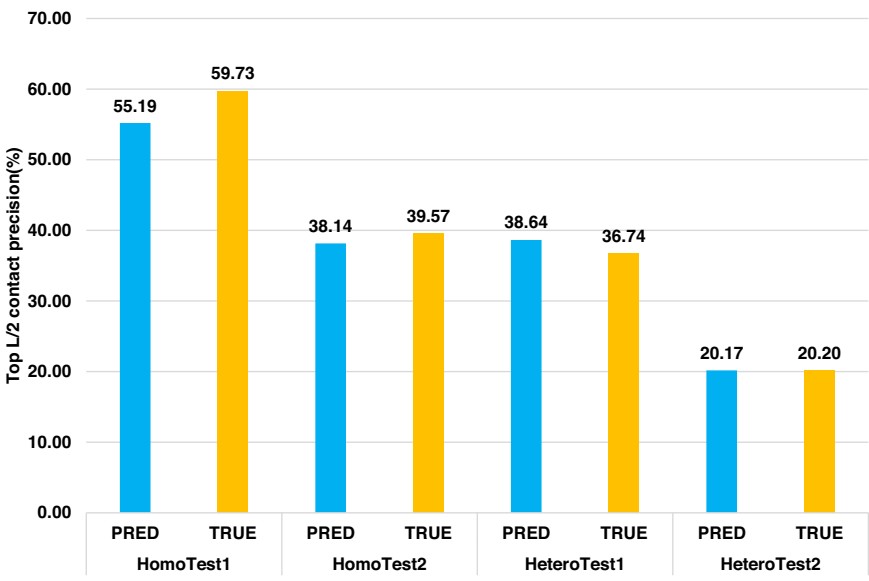

**Fig. 3 | Comparison of using AlphaFold-predicted tertiary structure (blue) and true tertiary structure (yellow) to generate intra-chain distance maps as input for predicting inter-chain distance maps on the four datasets.** Top L/2 contact prediction precision on the datasets is reported.

CDPred to leverage most tertiary structure information to predict inter-chain distances.

Moreover, we compared the top L/2 inter-chain contact prediction precision of using AlphaFold-predicted tertiary structures of monomers as input and using true tertiary structures of monomers in the bound state as input on the four datasets (Fig. 3). Using the true tertiary structures yields slightly better performance than using the AlphaFold-predicted structures on three out of four datasets (HomoTest1, HomoTest2, and HeteroTest2), but slightly worse performance on HeteroTest1. The $p$ value of the pairwise $t$-test of the difference on the four datasets is 0.6802, 0.8892, 0.9083, and 0.9963, respectively, indicating that the difference is not significant. The results show that the AlphaFold-predicted tertiary structures are sufficiently accurate for CDPred to make inter-chain distance prediction, even though using true tertiary structures as input can slightly improve the prediction accuracy overall. This is different from GLINTER whose accuracy of using true tertiary structures as input is substantially higher than using AlphaFold-predicted tertiary structures as input[21].

## High correlation between the precision of inter-chain contact predictions and predicted probability scores

The previous work on the intra-chain distance prediction[27] shows that the intra-chain distance prediction accuracy and predicted probability scores have a strong correlation, which can be used to select predicted intra-chain distance maps. Here, we investigate if a similar correlation exists in the inter-chain distance prediction. Figure 4 is a plot of the precision of top L/5 inter-chain contact predictions and the average of their probability scores for each target in the four test datasets. The correlation between the top L/5 inter-chain contact precision and the average predicted probability score is 0.7345. The high correlation suggests that the probability of inter-chain contacts predicted by CDPred can be used to estimate the confidence of the inter-chain prediction.

## The comparison between CDPred and AlphaFold2-multimer

AlphaFold2-multimer is currently the state-of-the-art method for predicting quaternary structures of multimers. To investigate if CDPred is complementary with AlphaFold2-multimer, we compare their inter-chain contact prediction accuracy on the four datasets. The comparison is not completely fair because the redundancy between the test datasets and AlphaFold2-multimer's training dataset is not removed.

We ran the latest version (Version 2) of AlphaFold2-multimer without templates to predict the quaternary structures for the dimers in the four test datasets. The inter-chain distance maps are extracted from the predicted quaternary structures. Each distance in the map is inverted to generate a contact probability map to be compared with the inter-chain contact map predicted by CDPred. Supplementary Figure 4 presents a target-by-target comparison of the top L/2 inter-chain contact prediction precision of CDPred and AlphaFold2-multimer for each target in the four test datasets. AlphaFold2-multimer has higher top L/2 precision than CDPred on the majority of the targets. However, for the very hard 44 targets on which the top L/2 precision of AlphaFold2-multimer is less than 10%, CDPred performs better than AlphaFold2-multimer on 15 targets, equally on 25 targets, and worse on 4 targets. On the 19 hard targets that the two methods perform differently, the average precision of CDpred is 14.8%, much higher than 1.79% of AlphaFold2-multimer. The $p$ value of the two-sided pairwise t-test of the difference is 0.0068, indicating it is significant. For instance, for target 7LB6, the top L/2 precision of CDPred is 44.62%, much higher than 0% of AlphaFold2-multimer. The Neff of the MSA of the target is 16.6. The results show that CDPred is complementary with AlphaFold2-multimer and can be particularly useful when the target is very hard and AlphFold2-multimer prediction has very low confidence. One possible application of CDPred is to use its predicted distance map to rank and select diverse quaternary structural models of hard targets predicted by AlphaFold2-multimer.

## An interesting inter-chain distance prediction example

Typically, when the MSA is shallow, the precision of inter-chain distance prediction is low due to the lack of information. However, CDPred still can accurately predict inter-chain distance for some targets with shallow MSAs. Figure 5 shows such a CASP13 homodimer target T0991, the distance map is visualized by matplotlib[28]. Its MSA has only one sequence. The TM-score[29] of the tertiary structure of the monomer of T0991 predicted by AlphaFold2 is 0.3104, indicating the predicted tertiary structure fold is not correct. However, the precision of top L/10, top L/5, and top L/2 inter-chain contacts derived from the distance map predicted by CDPred is 72.73, 68.18, and 56.36%, respectively, which is high. Figures 5a, b show the intra-chain distance maps of the AlphaFold-predicted tertiary structure and the true tertiary structure of the monomer, Fig. 5c shows the inter-chain contact map predicted by CDPred, and Fig. 5d the true inter-chain contact

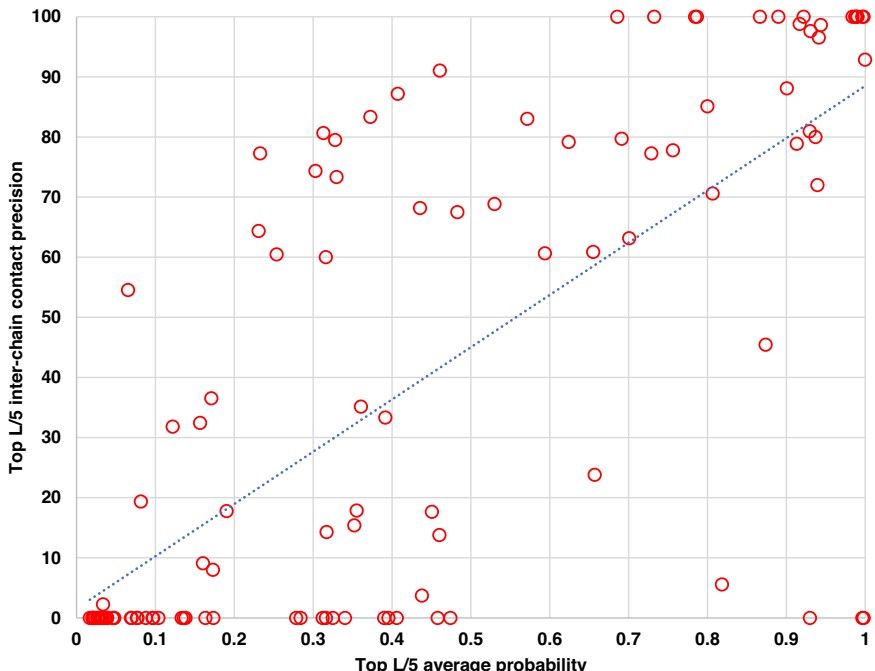

**Fig. 4 | The plot of inter-chain contact prediction precision against average contact probability.** The y-axis is the precision of top L/5 inter-chain contact predictions made by CDPred for a target, and the x-axis is the average probability of the top L/5 contact predictions for the target. Each point represents a dimer target in the four test datasets (HomoTest1, HomoTest2, HeteroTest1 and HeteroTest2).

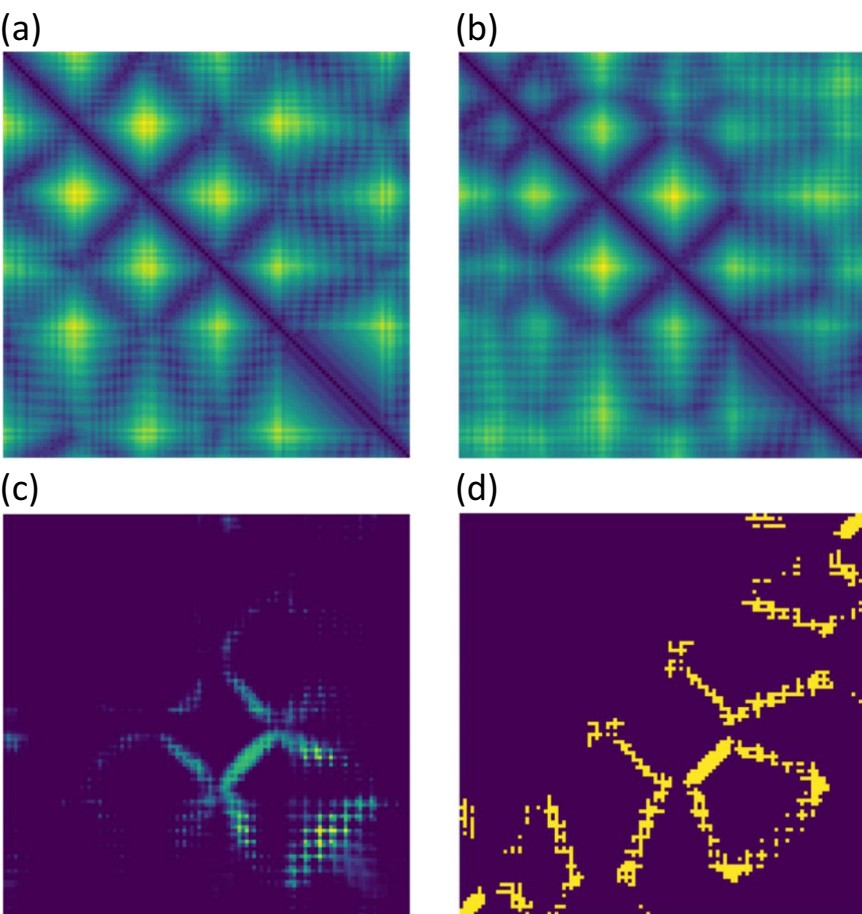

**Fig. 5 | The prediction for homodimer T0991 with a shallow MSA. a** The intra-chain distance map of the monomer predicted by AlphaFold. **b** The true intra-chain distance map of the monomer. **c** The inter-chain contact map predicted by CDPred. **d** the true inter-chain contact map.

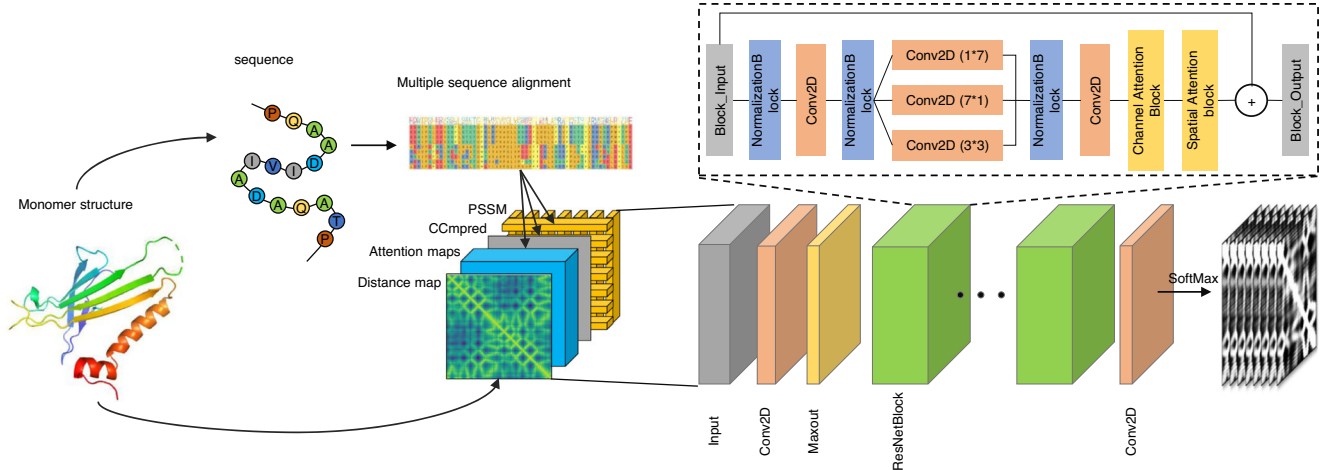

**Fig. 6 | Overview of the CDPred architecture.** CDPred simultaneously uses the tertiary structural information (i.e., intra-chain distance map of monomers), sequential information (PSSM), and residue-residue co-evolutionary information (i.e., co-evolutionary scores calculated by CCMpred and attention maps by MSA transformer) as input to predict inter-chain distance maps. The dimension of the input for the homomer dimer is $L \times L \times 186$ (L is the length of the monomer sequence), while the dimension of the input for the heterodimer is (L1 + L2) x

(L1 + L2) × 186 (L1 and L2 are the length of the two different monomers in the heterodimer). Each of the two output matrices has the same dimension as the input except for the number of output channels. The number of the output channels of the output layer is 42, storing the predicted probability of the distance in 42 distance bins. Two output matrices are generated, representing the two kinds of predicted inter-chain distance maps.

map. The predicted inter-chain contact map accurately recalls a large portion of the true inter-chain contacts.

## Methods

### Attention-based neural network architecture

Figure 6 illustrates the overall architecture of CDPred based on the channel-wise and spatial-wise attention mechanisms. CDPred takes the tertiary structures of monomers of a dimer as input and extracts the monomer sequences and intra-chain distance maps. For homodimers, since the sequences of the two monomers of a homodimer are the same, only one monomer tertiary structure is used as input. The monomer sequences are used to search the protein sequence databases to generate MSAs of dimers, which are used to generate residue-residue co-evolutionary scores, row attention maps, and position-specific scoring matrix (PSSM) as input features (see Features Subsection 4.2 for details). The complete input for CDPred is the concatenation of all the input features.

The input features stored in 2D tensors of multiple channels are first transformed by a 2D convolutional layer, followed by a Maxout layer[30] to reduce the dimensionality. The output of the Maxout layer is used as input for a series of deep residual network blocks empowered by the attention mechanism. The residual network has been widely used in computer vision and protein intra-chain distance and contact prediction[5,7,31]. Here, we combine the residual connection with other useful components to construct a residual block, which includes the normalization block (called RCIN) consisting of a row normalization layer (RN), column normalization layer (IN)[32], and instance normalization (IN)[33] for normalizing the feature maps, a channel attention squeeze-and-excitation (SE) block[34] for capturing the important information of different feature channels, and a spatial attention block[35] that captures signals between residues right after the channel attention block. Following the residual blocks, a 2D convolutional layer with the softmax function is used to classify the distance between any two residues from two monomers in a dimer into 42 distance bins (i.e., 40 bins from 2 to 22 Å with a bin size of 0.5 Å, plus a 0–2 Å bin and a >22 Å bin). Two kinds of inter-chain residue-residue distance are predicted at the same time: (1) the distance between the two closest heavy atoms from two residues used by most existing works in the field and (2) the $C_b$-$C_b$ distance between two residues used

by some recent works[36], resulting in two kinds of distance maps predicted.

### Features

The input features of CDPred contain (1) the tertiary structure information of monomers in the form of an intra-chain distance map, (2) pairwise co-evolutionary features, and (3) sequential amino acid conservation features, which are stored in an $L \times L \times N$ tensor (L is the length of the sequence of a monomer for a homodimer or the sum of the length of two monomers (L1 + L2) for a heterodimer). N is the number of feature channels for each pair of residues.

**Tertiary structure information of monomers.** The protein tertiary structure information of a monomer in a dimer is represented as an intra-chain distance map storing the distance between $C_b$ atoms of two residues in the monomer. For a homodimer, an intra-chain distance map ($L \times L \times 1$) computed from the tertiary structure of only one monomer is used. For a heterodimer, two intra-chain distance maps ($L1 \times L1 \times 1$ and $L2 \times L2 \times 1$) of the two monomers in the heterodimer are computed from their tertiary structures and added as the top left submatrix and the bottom right submatrix of the input distance map of the dimer of $(L1 + L2) \times (L1 + L2) \times 1$ dimension. The values of the other area of the input distance map of the heterodimer are set to 0. In the training phase, the true tertiary structures of monomers in the dimers are used to compute the intra-chain distance maps above. During the test/prediction phase, the tertiary structures of monomers predicted by AlphaFold are used to generate the intra-chain distance maps as input. Using predicted tertiary structures as input is more challenging but can more objectively evaluate the performance of inter-chain distance prediction because, in most situations, the true tertiary structures of the monomers are not known. A predicted tertiary structure also corresponds to an unbound tertiary structure, a term commonly used in the protein docking field.

**Co-evolutionary features.** MSAs are generated for homodimers or heterodimers as input for the calculation of their co-evolutionary features. To challenge the deep learning method to effectively predict inter-chain distance maps from noisy inputs, in the training phase, we use less sensitive tools or smaller sequence databases to generate

MSAs, but in the test phase, we use state-of-the-art tools and larger databases to generate the requisite MSAs. Specifically, in the training phase, for a homodimer, we use PSI-BLAST[37] to search the sequence of a monomer against Uniref90 (2018-04)[38] to generate the MSAs, and for a heterodimer, we follow the procedure in FoldDock[39] using the HHblits[40] to search against Uniclust30 (2017-10) to generate the MSA for each of the two monomers and then pair the two MSAs to produce an MSA for the heterodimer according to the organism taxonomy ID of the sequences.

In the test stage, for a homodimer, we use HHblits to search the sequence of a monomer against the Big Fantastic Database (BFD)[41] and Uniclust30 (2017-10), respectively, to generate two MSAs for a single chain of the homodimers, which are used to generate input features separately to make two predictions that are averaged as the final predicted distance map; for a heterodimer, an MSA is generated by the same procedure used in EvComplex2[42], which applies the jackhammer to search against Uniref90 (2018-04) to generate one MSA for each of the two monomers and then pairs the sequences from the two MSAs to produce an MSA for the heterodimer according to the highest sequence identity with the monomer sequences in each species. The MSA for a homodimer or a heterodimer is used by a statistical optimization tool CCMpred[43] to generate a residue-residue co-evolutionary score matrix ($L \times L \times 1$) as features and by a deep learning tool MSA transformer[22] to generate residue-residue relationship (attention) matrices ($L \times L \times 144$) as features. L is the number of columns in MSA.

**Sequential features.** The sequence profile (i.e., position-specific scoring matrix (PSSM)) of the protein generated by the PSI-BLAST search above contains the residue conservation information. The PSSM of a monomer in a homodimer or the vertical concatenation of two PSSMs of two monomers in a heterodimer in the shape of $L \times 20$ is tiled (i.e., cross-concatenated element by element) to generate sequential features of dimensionality $L \times L \times 40$.

### Training procedure and hyperparameters
The deep neural network uses the input features above to predict a heavy atom distance map and a $C_b$ distance map of shape $L \times L \times 42$. The 42 channels store the probability of a distance between two residues in 42 distance bins. The predicted inter-chain distance maps are compared with their true counterparts to calculate the cross-entropy loss to adjust the weights during training. For a heterodimer (L = L1 + L2), an output distance map of dimension $(L1 + L2) \times (L1 + L2) \times 42$ contains both inter-chain distance predictions and intra-chain distance predictions. Only inter-chain distance predictions are used to calculate the cross-entropy loss to train the networks, while the intra-chain distance predictions are ignored. The number of convolutional layers of CDPred is set to 156, and the number of filters of each convolutional layer is set to 64. The batch size in training is set to 1 due to the limitation of GPU memory. We used the Adam optimizer with a 1e-3 learning rate to train the model for the first 30 epochs to achieve fast convergence and used stochastic gradient descent with the 1e-4 starting learning rate and 10-time reduction every 20 epochs for the remaining 50 epochs to further reduce the training loss.

### Datasets and evaluation metrics
We use the DeepHomo training dataset[19] to train the homodimer inter-chain distance predictor. The whole dataset includes 4,132 homodimeric proteins with C2 symmetry. And after removing proteins that have > =30% sequence identity with the blind test datasets (HomoTest1 and HomoTest2) consisting of the targets of the CASP/CAPRI experiments using MMseq2[44], 4129 homodimers are left as training, validation, and internal test data. The same as DeepHomo, we select 300 of them as the validation data and 300 as the internal test data and use the rest as the training data. The test dataset used by DeepHomo

contains 28 targets collected from the CASP10-13 experiments is used as one blind homodimer test dataset (**HomoTest1**). Another test dataset used by GLINTER[21], which includes 23 homodimer targets collected from the CASP13 and 14, is used as the other blind homodimer test dataset (**HomoTest2**). The two blind test datasets have six common targets.

For heterodimers, we use the heterodimers in Apoc[45] to create the training, validation, and internal test datasets. After filtering out similar sequences at the 40% sequence identity threshold and removing the sequences with ≥30% sequence identity with the blind test datasets (HeteroTest1 and HeteroTest2), 3955 heterodimers are left. We randomly select 3576 of them as the training data, 198 as the validation data, and 181 as the internal test data. The test dataset used by GLINTER which contains nine heterodimer targets from the CASP13 and CASP14 experiments in conjunction with the CAPRI experiments is used as a blind test dataset (**HeteroTest1**). To create a larger blind test dataset, we collect the heterodimer released between 09-2021 and 11-2021 in the PDB. After filtering out similar sequences at a 40% sequence identity threshold and excluding sequences with >1000 residues targets, 55 heterodimers are left to create another blind test dataset (**HeteroTest2**).

Since the external methods, GLINTER and DeepHomo predict inter-chain contacts instead of inter-chain distances, to fairly compare CDPred with them, we use the precision of contact prediction as the evaluation metric. Specifically, the precision of top 5, 10, L/10 (or Ls/10), L/5 (or Ls/5), L/2 (or Ls/2), and L (or Ls) contact predictions (L: length of a monomer in homodimers, Ls: length of the shorter monomer in heterodimers) is computed and compared. A similar metric is also widely used in evaluating intra-chain contact prediction. Because DeepHomo and GLINTER predict inter-chain contacts at an 8 Å threshold, we use the same threshold to convert the distance maps predicted by CDPred into the binary contact map. A predicted inter-chain contact is correct if the minimal distance between the heavy atoms of the two residues is less than 8 Å. The accuracy order, accuracy rate[46], and AUC score are also used to evaluate the inter-chain distance prediction of CDPred. The accuracy order is the rank of the first correct contact prediction divided by the total number of residues of a dimer. AccRate is the percentage of dimers for which at least one of the top 10 inter-chain contact predictions is correct.

### Reporting summary
Further information on research design is available in the Nature Portfolio Reporting Summary linked to this article.

## Data availability
The test data generated in this study have been deposited in the Zenodo database under Creative Commons Attribution 4.0 International Public License at https://zenodo.org/record/6647564. The raw protein dimer data used in this study are available under the CC0 1.0 Universal (CC0 1.0) Public Domain Dedication at https://www.rcsb.org/. The source data generated in this study are provided in the Source Data file. The Uniclust30(2017-10) database used in this study is available under the Creative Commons Attribution-ShareAlike 4.0 International License at https://wwwuser.gwdg.de/~compbiol/uniclust/2017_10/. The Uniref90(2018-10) database used in this study is available under the Creative Commons Attribution 4.0 International (CC BY 4.0) License at https://www.uniprot.org/help/uniref. And the Big Fantastic Database (BFD) database used in this study is available at https://bfd.mmseqs.com/. Source data are provided with this paper.

## Code availability
The code of CDPred[47] is available at: https://github.com/BioinfoMachineLearning/CDPred.

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

## Acknowledgements

The work was partly supported by the Department of Energy, USA (grant #: DE-AR0001213 (J.C.), DE-SC0020400 (J.C.), and DE-SC0021303 (J.S. and J.C.)), National Science Foundation (grant #: DBI1759934 (J.C.) and IIS1763246 (J.C.)), and National Institutes of Health (grant #: R01GM093123 (J.C.), R01GM146340 (J.C.), and R35GM118039 (J.S.)).

## Author contributions

J.C. conceived the project. Z.G. and J.C. designed the experiment. Z.G. performed the experiment and collected the data. Z.G., J.C., J.L., and J.S. analyzed the data. J.L. and J.S. provided some datasets. Z.G. and J.C. wrote the manuscript. Z.G., J.C., and J.S. edited the manuscript.

## Competing interests

The authors declare no competing interests.
