## [Peer Review File · Nature Communications]

Prediction of inter-chain distance maps of protein complexes with 2D attention-based deep neural networksREVIEWER COMMENTS

Reviewer #1 (Remarks to the Author):

In this manuscript, the authors have developed a protein-protein contact prediction method for homodimers and heterodimers through deep neural networks with transformer row attention features and two new modules including channel-wise and spatial-wise attention mechanisms. The improvement over previous similar models is significant and the manuscript is well written. The valuable part of the present model is the improvement in the architecture of its networks. However, the ultimate goal of protein-protein contact predictions is to obtain the accurate structures of corresponding protein-protein complexes, which is not addressed in this study. Regarding the model and results, here are some comments or questions that may be considered.

Major concerns:

- 1) One interesting part of the model is that the authors combined the predictions of CDPred from two kinds of MSAs to improve the prediction accuracy. Specifically, the authors predicted the protein complex distance map by two different MSAs on BFD and Uniclut30 databases. It would be necessary to analyze the differences between the two MSAs to show the rationality for the improvement of the accuracy. In addition, how accurate is the prediction by directly combining the two MSAs? It would also be valuable to show and discuss the results based on the MSA from BFD or Uniclust30 alone in Section 2.2.
- 2) The authors used two different orders of monomer A and monomer B (AB and BA) in each heterodimer to generate input features for CDPred. The average of the Outputs for the two orders is used as the final prediction. How did the authors average the results? Because of the change of orders, it is not reasonable to average the prediction directly. To avoid confusion, the authors may need to specify how to average the predictions. In addition, the authors should analyze the influence of the type of orders on the precision.
- 3) The authors mentioned that "The NoContact intra-chain distance maps work even slightly better for inter-chain distance map for homodimers than the FullMap intra-chain distance map, but they perform worse for heterodimers." How did the authors only keep contact/non-contact intra-chain distance information? Did the authors ignore the rest of distance information? If so, it seems to be not reasonable, since the diagonal distance information is already zero in the intra-chain distance map. In addition, the NoContact intra-chain distance information obtained different results on homodimer and heterodimers. Does keeping only non-contact intra-chain distance information produce positive effects?
- 4) The authors only list the precisions with predicted structures by AlphaFold2 in Tables 1, 2, 3, and 4. In addition, the authors showed the top L/2 precision for the bound structures in Figure 4. We noticed that the top L/2 precisions for the bound and predicted structures are almost the same in HomoTest1 and HeteroTest2. Generally, the quality of monomer structures would greatly affect the precision of predicted contact maps. The authors should list all the precisions of bound structures in Tables 1, 2, 3, and 4 for comparison and discuss the corresponding impact of structure quality. In addition, the precision for predicted structures outperform the precision for bound structures by 12 percent on HeteroTest1. That is strange because the bound structures are definitely more accurate than the predicted. One possible reason could be an overfitting of their CDPred model to the predicted structures.
- 5) "Two kinds of inter-chain residue-residue distance are predicted at the same time". The authors predict two kinds of inter-chain distance maps. However, the authors did not evaluate and analyze the two kinds of predictions, nor did they indicate which prediction was used in the result sections. In addition, GLINTER predicts the contacts between the two closest heavy atoms from two monomers in a dimer.

6) Another major concern is that the CDPred cannot predict the corresponding structures of protein-protein complexes, which is expected to be the ultimate goal of protein-protein contact predictions. The authors may need to demonstrate the values of their CDPred method, compared with the state-of-the-art approaches like AF2Complex and AlphaFold-Multimer for modeling protein-protein complex structures.

Minor issues:

1) "the tertiary structure of the monomer of T10991"

The word "T10991" is a typo, it should be "T0991".

2) What are the exact hyper-parameters of CDPred?

Reviewer #2 (Remarks to the Author):

In the draft, the authors provide a novel sequence-based model that can predict the tertiary structures of protein monomers or the quaternary structures of protein complexes from residue-residue distance information. The results are significant in the field with a great improvement in predictive performance. It meets the expected standards. However, it has several points that need to be addressed.

1. The method used to evaluate output should be added. In the manuscript, it applies precision to evaluate the performance of the model, which is necessary. However, all the results are based only on precision, which seems not strong enough. It may be necessary to include other evaluation scores, such as accuracy rate, accuracy order, and AUC (since the model provides scores)...

2. Comparing with additional methods, such as DNCON2_inter, DeepComplex, and ComplexContact.

3. For the monomer and shorter monomer, both abbreviations are L. It's better to use a unique symbol for different representatives.

4. In figure 1, it provides two subfigures with the same number; I wonder if it is possible?

Point-by-point response to the review comments

Response to Reviewer 1

1. One interesting part of the model is that the authors combined the predictions of CDPred from two kinds of MSAs to improve the prediction accuracy. Specifically, the authors predicted the protein complex distance map by two different MSAs on BFD and Uniclust30 databases. It would be necessary to analyze the differences between the two MSAs to show the rationality for the improvement of the accuracy. In addition, how accurate is the prediction by directly combining the two MSAs? It would also be valuable to show and discuss the results based on the MSA from BFD or Uniclust30 alone in Section 2.2.

Thank you for the great comments. According to your suggestion, we add a new Section 2.2 to analyze the differences between the two kinds of MSAs, the contact prediction accuracy based on each of them, the performance of averaging the contact maps predicted, and the accuracy of directly combining the two kinds of MSAs as input on the two homodimer test datasets. We report the number of sequences (N) and the number of effective sequences (Neff) of each MSA of each target in HomoTest1 and HomoTest2 as well as the top L/2 inter-chain contact prediction precision of the four methods (CDPred_BFD of using BFD MSA as input, CDPred_Uniclust of using Uniclust MSA as input, CDPred_ComMSA of combining the two MSAs into one MSA as input, and CDPred of averaging the distance maps of CDPred_BFD and CDPred_Uniclust) in supplemental **Table S1** and **Table S2**. The Neff and contact prediction precision for CDPred_BFD and CDPred_Uniclust vary from target to target. The Pearson correlation coefficient between the difference of Neff and the difference of the top L/2 contact precision between the CDPred_BFD and CDPred_Uniclust is 0.31 and 0.67 for HomoTest1 and HomoTest2, respectively. The moderate correlation indicates that the depth of multiple sequence alignment has a positive impact on the prediction accuracy.

CDPred_ComMSA which combines the two MSAs to generate a deeper MSA as input performs better than both CDPred_BFD and CDPred_Uniclust on HomoTest1. It performs better than CDPred_BFD but worse than CDPred_Uniclust on HomoTest2. The results suggest that directly combining two MSAs can be beneficial even though it is not guaranteed that the combination will always yield the best performance.

CDPred performs slightly better than CDPred_ComMSA in terms of the top L/2 prediction precision on both datasets (55.19% versus 55.13% on HomoTest1 and 38.14% versus 36.14% on HomoTest2) indicating that averaging the distance maps predicted from the two MSAs is more effective than combining the two MSAs as input. CDPred also performs better than CDPred_BFD and CDPred_Uniclust, indicating that combining the distance maps predicted from the two MSAs can reduce the variance of the two distance maps and smooth the prediction to improve the prediction accuracy.

2. The authors used two different orders of monomer A and monomer B (AB and BA) in each heterodimer to generate input features for CDPred. The average of the Outputs for the two orders is used as the final prediction. How did the authors average the results? Because of the change of orders, it is not reasonable to

average the prediction directly. To avoid confusion, the authors may need to specify how to average the predictions. In addition, the authors should analyze the influence of the type of orders on the precision.

Thanks for the great comment. We added a flowchart of how to average the two orders in supplemental **Figure S1** to explain how the distance maps predicted from the two orders are combined. We also provide a target by target comparison of the two orders in terms of Ls/5 contact prediction precision on HeteroTest1 and HeteroTest2 datasets in supplemental **Figure S2** (Ls: the length of the shorter monomer unit in a heterodimer). The evaluation of contact prediction on the HeteroTest1 and HeteroTest2 for two different orders CDPred(A_B), CDPred(B_A) and their average CDPred is shown in supplemental **Table S3** and **Table S4**.

The accuracy of CDPred(A_B) and CDPred(B_A) varies from target to target and from dataset to dataset. Sometimes the precision of the two orders can be substantially different (see **Figure S2** for some examples). However, a pairwise t-test shows that there is no significant difference between the two orders. Even though averaging the contact maps predicted in two different orders does not always yield the best accuracy, it makes the performance more stable by reducing the variance and smoothing the prediction. For instance, CDPred often delivers either the best or medium prediction accuracy in comparison with CDPred(A_B) and CDPred(B_A). The results and analysis above have been added into the main manuscript.

3. The authors mentioned that "The NoContact intra-chain distance maps work even slightly better for inter-chain distance map for homodimers than the FullMap intra-chain distance map, but they perform worse for heterodimers." How did the authors only keep contact/non-contact intra-chain distance information? Did the authors ignore the rest of distance information? If so, it seems to be not reasonable since the diagonal distance information is already zero in the intra-chain distance map. In addition, the NoContact intra-chain distance information obtained different results on homodimer and heterodimers. Does keeping only non-contact intra-chain distance information produce positive effects?

Thanks for pointing out the problem. Indeed, setting contact (or non-contact) intra-chain distances to 0 to ignore them can be problematic as the deep network can treat them like the 0 values on the diagonal. Therefore, we decided to remove the section of analyzing the impact of NonContact intra-chain distances or contact intra-chain distances on the inter-chain distance prediction. This removal does not affect any other part of the paper because all the other results are obtained using the full intra-chain distance map as input.

4. The authors only list the precisions with predicted structures by AlphaFold2 in Tables 1, 2, 3, and 4. In addition, the authors showed the top L/2 precision for the bound structures in Figure 4. We noticed that the top L/2 precisions for the bound and predicted structures are almost the same in HomoTest1 and HeteroTest2. Generally, the quality of monomer structures would greatly affect the precision of predicted contact maps. The authors should list all the precisions of bound structures in Tables 1, 2, 3, and 4 for comparison and discuss the corresponding impact of structure quality. In addition, the precision for predicted structures outperform the precision for bound structures by 12 percent on HeteroTest1. That is strange because the bound structures are definitely more accurate than the predicted. One possible reason could be an overfitting of their CDPred model to the predicted structures.

Thank you for the insightful comment. According to your suggestions, we add a new Section 2.5 to investigate the impact of the quality of the tertiary structure input on inter-chain contact prediction (see Section 2.5). We reanalyzed the predictions using the AlphaFold-predicted tertiary structures and true bound tertiary structures as input. We also fixed an error in using the true bound tertiary structure as input to predict the distance map for a target H1017 in the HeteroTest1 dataset, which was caused by an uncommon amino acid type in the PDB file of its true tertiary structure leading to an incorrectly computed intra-chain distance map. Before the bug was fixed, the TopL/2 contact precision of using the true tertiary structure as input for H1017 is 0% due to the incorrect calculation of the intra-chain distance map. After the bug is fixed, this precision is 72.72%. Fixing this error substantially increases the average prediction accuracy when using the true bound tertiary structure as input on the dataset. Based on the corrected results, we compare the performance of using the predicted and true tertiary structure inputs using the final average predictor CDPred instead of using CDpred_BFD (using BFD MSA only) or CDPred_Uniclust (using UniclustMSA only).

The TM-scores of the tertiary structure of each monomer unit of each dimer and the contact prediction precision of CDPred on the four datasets (HomoTest1, HomoTest2, HeteroTest1, and HeteroTest2) are reported in new supplemental **Table S5**, **Table S6**, **Table S7**, and **Table S8**, respectively. The average TM-scores of the predicted tertiary structures for HomoTest1 and HomoTest2 are 0.95 and 0.90, for Chain A of heterodimers in HeteroTest1 and HeteroTest2 are 0.90 and 0.89, and for Chain B of heterodimers in HeteroTest1 and HeteroTest2 are 0.95 and 0.88, respectively, indicating the AlphaFold predicted tertiary structures have high quality. The Pearson's correlation between the TM-scores of the predicted tertiary structures and top L/2 contact prediction precision is 0.19. The weak correlation may be partly due to that the quality of predicted tertiary structures is high enough in general for CDPred to leverage most tertiary structure information to predict inter-chain distances.

Moreover, we compared the top L/2 inter-chain contact prediction precision of using AlphaFold predicted tertiary structures of monomers as input with using the true tertiary structures of monomers in the bound state as input on the four datasets based on the corrected results (see details in **Figure 4**). Using the true tertiary structures yields slightly better performance than using the AlphaFold predicted structures on three out of four datasets (HomoTest1, HomoTest2 and HeteroTest2), but slightly worse performance on HeteroTest1. The p-value of the pair-wise t-test of the difference on the four datasets is 0.6802, 0.8892, 0.9083, and 0.9963, respectively, indicating that the difference is not significant. The results show that the AlphaFold-predicted tertiary structures are sufficiently accurate for CDPred to make inter-chain distance prediction, even though using true tertiary structures as input can slightly improve the prediction accuracy overall.

5. "Two kinds of inter-chain residue-residue distance are predicted at the same time". The authors predict two kinds of inter-chain distance maps. However, the authors did not evaluate and analyze the two kinds of predictions, nor did they indicate which prediction was used in the result sections. In addition, GLINTER predicts the contacts between the two closest heavy atoms from two monomers in a dimer.

Thank you for pointing out the issue. We found an error in the previous description of two kinds of inter-chain distance prediction "Cb-Cb distance between two residues used by

some recent works (e.g., GLINTER)." The citation is not correct as you pointed out. GLINTER uses the two closest heavy atoms of two residues from two monomers in the dimer to determine if they are in contact. The work that uses Cb-Cb distance is Soltanikazemi, Elham, et al. "Distance-based reconstruction of protein quaternary structures from inter-chain contacts." *Proteins: Structure, Function, and Bioinformatics* 90.3 (2022): 720-731. We have corrected the description in the manuscript. In this work, we focus on using the same definition as GLINTER and DeepHomo for inter-chain contact in order to compare them on the same standard. We use this distance maps based on this definition in all the evaluation. We have added two sentences in Section 2.1 to clarify the definition of the inter-chain contact map used in the results sections. The added two sentences are "The definition of inter-chain contact is the same as GLINTER and DeepHomo, i.e., a pair of inter-chain residues is considered a contact if the distance between their two closest heavy atoms less than 8 Å. This definition is used to evaluate all the inter-chain contact predictions in this work. "

6. Another major concern is that the CDPred cannot predict the corresponding structures of protein-protein complexes, which is expected to be the ultimate goal of protein-protein contact predictions. The authors may need to demonstrate the values of their CDPred method, compared with the state-of-the-art approaches like AF2Complex and AlphaFold-Multimer for modeling protein-protein complex structures.

Thanks for your great points. We added a new Section 2.7 to compare CDPred and the latest version of AlphaFold2-multimer to identify the unique value of CDPred. We compare their inter-chain contact prediction accuracy on the four datasets. The comparison is not completely fair because the redundancy between the test datasets and AlphaFold2-multimer's training dataset is not removed. We ran the latest version (version 2) of AlphaFold2-multimer without templates to predict the quaternary structures for the dimers in the four test datasets. The inter-chain distance maps are extracted from the predicted quaternary structures. Each distance in the map is inverted to generate a contact probability map to be compared with the inter-chain contact map predicted by CDPred. Supplemental **Figure S3** presents the target-by-target comparison of top L/2 inter-chain contact prediction precision of CDPred and AlphaFold2-multimer for each target in the four test datasets. AlphaFold2-multimer has a higher top L/2 precision than CDPred on the majority of the targets. However, for the very hard 44 targets on which the top L/2 precision of AlphaFold2-multimer is less than 10%, CDPred performs better than AlphaFold2-multimer on 15 targets, equally on 25 targets, and worse on 4 targets. On the 19 targets that the methods perform differently, the average precision of CDPred is 14.8%, much higher than 1.79% of AlphaFold2-multimer. The p-value of the pairwise t-test of the difference is 0.0068, indicating it is significant. For instance, on 7LB6 target, the top L/2 precision of CDPred is 44.62%, much higher than 0% of AlphaFold2-multimer. The Neff of the MSA of the target is 16.6. The results show that CDPred is complementary with AlphaFold2-multimer and can be particularly useful when the target is very hard and AlphaFold2-multimer prediction has very low confidence. One possible application of CDPred is to use its predicted distance map to rank and select diverse quaternary structures of hard targets predicted by AlphaFold2-multimer.

7. "the tertiary structure of the monomer of T10991" The word "T10991" is a typo, it should be "T0991".

Thank you for pointing this out. We have it corrected.

8. What are the exact hyper-parameters of CDPred?

Thanks for the great comment. We add the hyper-parameter information of CDPred into Section 4.3.

Response to Reviewer 2

1. The method used to evaluate output should be added. In the manuscript, it applies precision to evaluate the performance of the model, which is necessary. However, all the results are based only on precision, which seems not strong enough. It may be necessary to include other evaluation scores, such as accuracy rate, accuracy order, and AUC (since the model provides scores)...

Thanks for your great suggestion. In the result section, we add the evaluation results on the four test datasets in terms of accuracy rate, accuracy order, and AUC score (see the detailed new results in Tables 1, 2, 3, and 4). Consistent with the results based on the contact prediction precision, CDPred performs substantially better than DNCON2_inter, DeepComplex, DeepHomo, and GLINTER in most cases in terms of Accuracy Order, Accuracy Rate, and AUC score. The discussions of the new results are added into Sections 2.1 and 2.3. We also add the definition of these metrics into Section 2.1 and Section 4.4. We cite the paper originally defining accurate rate and accurate order.

2. Comparing with additional methods, such as DNCON2_inter, DeepComplex, and ComplexContact.

Thank you for another great suggestion. We add the comparison with DNCON2_inter and DeepComplex on HomoTest1 and HomoTest2 into Table 1 and Table 2 and the comparison with DeepComplex on HeteroTest1 and HeteroTest2 into Table 3 and Table 4. Unfortunately, we cannot compare CDPred with ComplexContact because its web server is down and there is no other way to obtain its results on the four test datasets used in this work. However, as ComplexContact was developed by the same group who developed the latest state-of-the-art method GLINTER and was several years older and much less accurate than GLINTER, we expect that CDPred performs better than ComplexContact because CDPred performs substantially better than GLINTER on the benchmark. To recognize the historical contribution of ComplexContact, we add a citation to it in the Introduction Section.

3. For the monomer and shorter monomer, both abbreviations are L. It's better to use a unique symbol for different representatives.

Thank you for the great comment. Now, we use Ls to represent the length of the shorter monomer in heterodimers, while using L to denote the length of the monomer in the homodimers. This change is applied to the entire article.

4. In figure 1, it provides two subfigures with the same number; I wonder if it is possible?

Thank you for pointing out the problem. The first of these two subfigures is redundant except it uses a different color. Now it is removed.

REVIEWERS' COMMENTS

Reviewer #2 (Remarks to the Author):

The current version of the paper is ready to publish. The authors provide enough results based on my comments, and enough data to support their conclusion. Thank you.

Response to Review Comments

Reviewer #2 (Remarks to the Author):

The current version of the paper is ready to publish. The authors provide enough results based on my comments, and enough data to support their conclusion.

Response:

Thank you very much.